# Computing water flow through complex landscapes, Part 1: Incorporating depressions in flow routing using FlowFill

Kerry L. Callaghan[1] and Andrew D. Wickert[1,2]

[1]Department of Earth Sciences, University of Minnesota, Minneapolis, MN, USA
[2]Saint Anthony Falls Laboratory, University of Minnesota, Minneapolis, MN, USA

*Correspondence to:* Kerry Callaghan (calla350@umn.edu)

**Abstract.** Calculating flow routing across a landscape is a routine process in geomorphology, hydrology, planetary science, and soil and water conservation. Flow-routing calculations often require a preprocessing step to remove depressions from a DEM to create a 'flow-routing surface' that can host a continuous, integrated drainage network. However, real landscapes contain natural depressions that trap water. These are an important part of the hydrologic system, and should be represented in flow-routing surfaces. Historically, depressions (or 'pits') in DEMs have been viewed as data errors, but the rapid expansion of high-resolution, high-precision DEM coverage increases the likelihood that depressions are real-world features. To address this longstanding problem of emerging significance, we developed FlowFill, an algorithm that routes a prescribed amount of runoff across the surface in order to flood depressions, but only if enough water is available. This mass-conserving approach typically floods smaller depressions and those in wet areas, integrating drainage across them, while permitting internal drainage and disruptions to hydrologic connectivity. We present results from two sample study areas to which we apply a range of uniform initial runoff depths and report the resulting filled and unfilled depressions, the drainage network structure, and the required compute time. For the reach- to watershed-scale examples that we ran, FlowFill compute times ranged from approximately 1 to 30 minutes, with compute times per cell of 0.0001 to 0.006 s.

## 1 Introduction

Flow routing based on Digital Elevation Models (DEMs) determines the paths taken by surface water (absent human interventions) and its associated sediment and/or dissolved load. Flow routing algorithms are applied across a broad range of fields, including hydrologic and geomorphic modeling, topographic analysis, planetary science, and palaeoclimate. They are a critical component of both hydrologic (Neal et al., 2011; Ng et al., 2018; Ray et al., 2016) and geomorphic (Adams et al., 2017; Coulthard et al., 2013) models, with the former including watershed-scale processes and flood risk. In the latter case, flow routing is often recomputed over time to simulate the feedback between evolving topography and drainage patterns (Hobley et al., 2017; Tucker et al., 2011). Flow-routing calculations and drainage-network construction also form the basis for topographic analysis algorithms to automatically pick channel heads (Clubb et al., 2014; Passalacqua et al., 2010; Pelletier, 2013), segment watersheds into representative hydrological units (Czuba and Foufoula-Georgiou, 2014; Ng et al., 2018; Teng et al., 2017), and link river-channel form with rates of tectonic uplift (Duvall et al., 2004; Perron and Royden, 2013; Willgoose et al.,

1991) or subsidence (Paola et al., 1992; Wickert and Schildgen, 2019). These same tools have been applied to understand valley networks on Mars (Luo and Stepinski, 2009; Molloy and Stepinski, 2007), the impacts of freshwater forcing on climate during the most recent deglaciation (Ivanovic et al., 2017, 2018; Riddick et al., 2018), and links between palaeo-drainage networks and modern economic and agricultural resources (Craddock et al., 2010). Flow-routing algorithms are thus vital to our understanding of landscapes, climates, and water resources.

Multiple methods exist to distribute flow across the landscape, and these range from simple approaches to find the path of steepest descent from one DEM pixel to another (O'Callaghan and Mark, 1984) to full shallow-water equation solvers (McGuire et al., 2013). Simple topographically-driven flow routing approaches are the most popular because they are quick to compute (e.g., Braun and Willett, 2013; Gallant and Wilson, 1996; Schwanghart and Scherler, 2014), agnostic to the amount of rainfall or runoff applied, and applicable over length scales from puddles (e.g., Chu et al., 2013) and small catchments (e.g., Ng et al., 2018; Teng et al., 2017) to continents (e.g., Coe, 2000; Riddick et al., 2018; Wickert, 2016). It is this type of flow-routing algorithm that we consider for the remainder of this paper.

Because these simple algorithms route flow down the topographic gradient, local enclosed depressions in the landscape present a problem. Flow cannot be routed across them, so they disconnect the hydrologic network. Prior to computing flow routing, a DEM may be preprocessed to create a flow-routing surface in which depressions are managed in order to reliably extract stream networks (Metz et al., 2011). This step often results in the removal of all depressions from the original DEM (e.g., Jenson and Domingue, 1988; Martz and Garbrecht, 1998, 1999; O'Callaghan and Mark, 1984; Soille, 2004; Cordonnier et al., 2018). Removing depressions produces a topographic surface over which a flow-routing calculation will produce a fully connected hydrologic network.

The outright removal of all depressions, enforcing integrated drainage, means that we are biasing our landscape or hydrologic analyses towards what we are more easily able to calculate: an integrated drainage network within a continuous downhill-sloping topography. By building such a flow-routing surface, we selectively remove information about the complexity of the real landscape. Real hydrologic networks include both transport of water across the land surface and temporary storage of water in depressions.

We have developed a tool, FlowFill (Callaghan, 2019), that permits flow-routing calculations across landscapes that may contain real depressions. To do so, FlowFill employs mass-conserving and hydrologically consistent depression filling, allowing a user-selected depth of runoff to be spread across the landscape and flood only those depressions that would be filled by an overland flow event of the chosen magnitude. This approach eschews the assumption of fully-integrated drainage and can help to improve the fit between the computed hydrologic network and field conditions (Coe, 2000).

## 2 Background and Motivations

Most existing approaches to managing depressions in flow-routing calculations assume that these are data errors and should be removed (Lindsay and Creed, 2005). In a historical context, this was a reasonable assumption: many DEMs were constructed from sparse data, and small data errors may have been enough to produce depressions, especially in flat areas (O'Callaghan

and Mark, 1984). By filling these depressions, researchers bypassed this technical challenge to begin analysing the structure of drainage networks, and continue to make new discoveries based on this approach (e.g., Hooshyar et al., 2017; Seybold et al., 2017). However, all of these analyses rely an assumption of full drainage connectivity. This assumption may be broken in regions of lakes and basins, such as formerly-glaciated terrains (Lai and Anders, 2018) and regions in which tectonic deformation isolates individual internally drained basins between ranges (e.g. Ballato et al., 2017; Sobel et al., 2003). Such depressions occur on a sub-continental scale as well, especially in arid regions, and our lack of an approach to self-consistently include these in our flow-routing algorithms inhibits efforts to construct and analyse large-scale drainage patterns and their changes over time (e.g., Wickert, 2016).

Several methods have been developed to remove depressions from a DEM during creation of a flow-routing surface, or to otherwise connect drainage across the landscape. A popular choice is a flood-fill algorithm, in which the elevation of all cells in depressions are raised to the level of their outlets (e.g., Jenson and Domingue, 1988; Martz and Jong, 1988). Martz and Garbrecht (1998) presented an alternative approach to the flood fill, which they called 'breaching'. The breaching approach lowers select cells at depression outlets, reducing the amount by which depression cells need to be raised. Soille et al. (2003) extended this concept with their carving method. Rather than raising cells inside depressions, surrounding cells are lowered in order to eliminate all depressions in the topography. Combined methods both raise and lower cell elevations to minimize the topographic difference between the original DEM and the flow-routing surface (Lindsay and Creed, 2005; Schwanghart and Scherler, 2017; Soille, 2004). Grimaldi et al. (2007) proposed a physically-based method based on steady-state topography that adjusts the elevation of cells in a DEM to that of a continuous river long profile. Metz et al. (2011) sidestep the need for topographic adjustments by instead using a least-cost-path method to determine drainage paths; this allows water to flow uphill to escape depressions and is employed in the GRASS GIS 'r.watershed' algorithm (Neteler et al., 2012).

Each of the methods mentioned above ignores all depressions in the DEM, either by removing them or by allowing water to flow across them, and discounts the significant hydrologic impact of real depressions. In addition to helping to control drainage pathways (Govers et al., 2000), depressions set the volume of water that can pond on the land surface (Abedini et al., 2006; Hansen et al., 1999), thereby enhancing infiltration and both slowing and reducing surface runoff (Darboux and Huang, 2005). For example, in the prairie wetland region of North America, natural depressions hydrologically disconnect a landscape unless a runoff event is large enough to fill and overtop them (Arnold, 2010; Shaw et al., 2012). Furthermore, the growing availability of high-resolution and high-precision topographic data makes it increasingly difficult to support the assumption that these depressions are errors that should be removed (Arnold, 2010; Li et al., 2011; Lindsay and Creed, 2006). Even coarse-resolution data on a global scale can resolve real internally-drained basins (e.g. Riddick et al., 2018; Wickert, 2016). Thus, the degree to which the drainage is integrated and the flows are connected is not static, but rather varies as a function of runoff. All of these arguments motivate an approach that allows depressions to be filled in a way that is hydrologically realistic.

The aim of FlowFill is to generate flow-routing surfaces with an amount of drainage integration (and hence, hydrologic connectivity) that is appropriate for the amount of input runoff and the shape of the land surface. Here, we use 'drainage integration' to refer to the degree to which streams are connected (via lakes) instead of terminating in depressions, and greater drainage integration leads to a greater degree of surface-water hydrologic connectivity. Our goal in generating these surfaces

is not a new one: Martz and Garbrecht (1998) noted the then-unrealized importance of incorporating depression storage into derived drainage patterns. Appels et al. (2011) and Chu et al. (2013) investigated the role of microtopography in connecting small surface depressions (puddles), and Shaw et al. (2013) required that ponds be filled by rainfall in their contributing areas before they be allowed to spill over their boundaries and integrate into the remainder of the catchment. In our approach, we

developed a cell-by-cell runoff-routing algorithm that fills depressions while conserving runoff volume in real landscapes. This open-source algorithm, FlowFill (Callaghan, 2019), can compute flow-routing surfaces across a wide range of landscapes, and is applicable at a range of length scales (see Table 1).

## 3 Methods

We present an algorithm to create more realistic flow-routing surfaces by flooding depressions with mass-conserved surface

runoff. Depressions that are small or have large catchments become completely filled, allowing flow to cross them, whereas larger depressions may be only partially filled and continue to be hydrologic sinks (Figure 1). FlowFill works by applying a user-selected runoff depth across the landscape and moving water downslope. If a parcel of water encounters a depression, as much of that parcel that can be contained by the depression before it overflows into an adjoining pixel is left behind (Callaghan, 2019). This enables users to perform flow-routing across a landscape whose level of hydrologic connectivity changes through

time due to storms, seasonality, or changing climate. The process of creating this flow-routing surface is summarised in Figure 2.

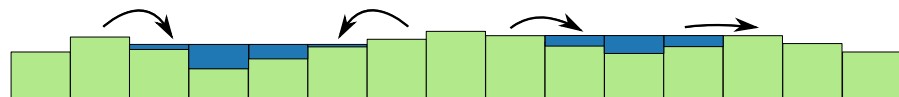

**Figure 1.** Filled depressions allow runoff to pass over them while unfilled depressions act as sinks to flow. The green blocks indicate the topographic surface with variable elevation, and the blue indicates the final water depth after running FlowFill. The lake on the left is not completely filled, and its level therefore is lower than the height of cells on either side. Water would continue to flow into this depression from all directions. On the right, water has completely filled a depression. Any flow entering this area from the left is able to flow out on the right and continue downslope.

FlowFill produces a flow-routing surface through an unconditionally stable method that iteratively routes water from cell to cell across the domain (Figure 3). The required inputs are a DEM, a user-selected starting runoff value, and a user-selected threshold for convergence. This threshold specifies the maximum amount of water that may be moved between cells between

two adjacent iterations for that iteration to count towards the eventual completion of the FlowFill calculation.

In Flowfill, water moves downslope, moving water from each cell in the domain once per iteration. The downstream direction for water movement is defined as the steepest downslope direction using a D8 approach (i.e. through comparison between the elevation of a 'target cell' and the elevation of the eight neighbours with which it shares either an edge or a corner). In cases

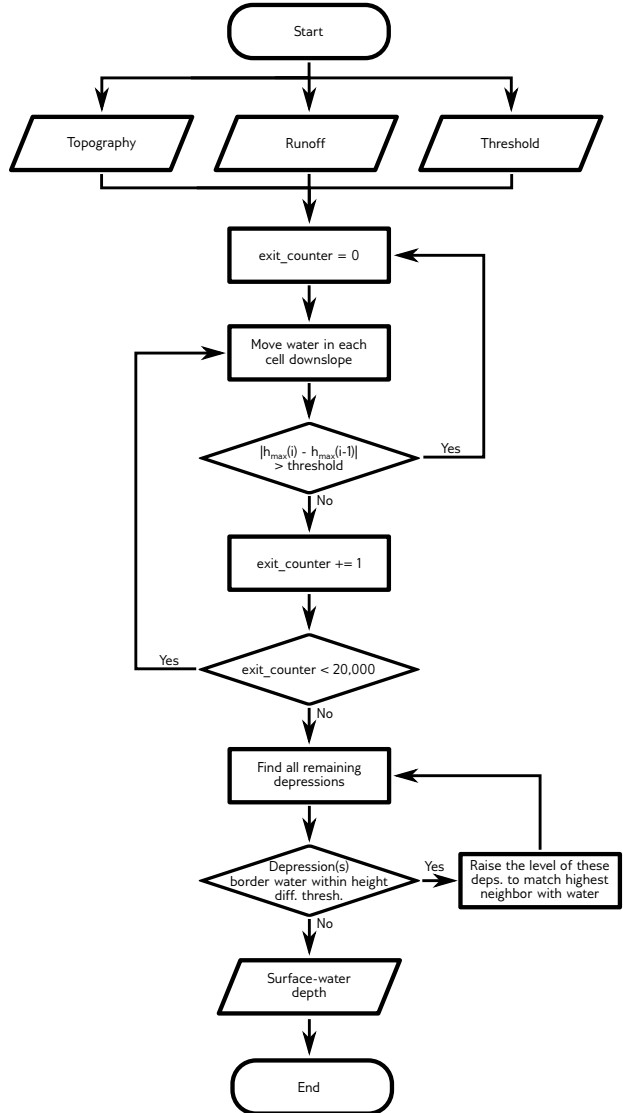

**Figure 2.** FlowFill flowchart. '$h_{max}$' is the maximum amount of water moved from a single cell to another during each iteration ('$i$'). The majority of the runtime is spent moving water from each target cell to those below.

in which two or more directions tie for being the steepest downslope direction, the user selects whether a preferential or a random direction is preferred. We provide this choice since selection of a preferential direction may systematically impact the ultimate destination of the water, whereas a random direction will solve this problem to some extent but make the result non-deterministic. When a preferential direction is selected, we arbitrarily route water preferentially Northwest, then West, Southwest, South, and continue anti-clockwise with the least preferred direction being North. These cases should be rare since

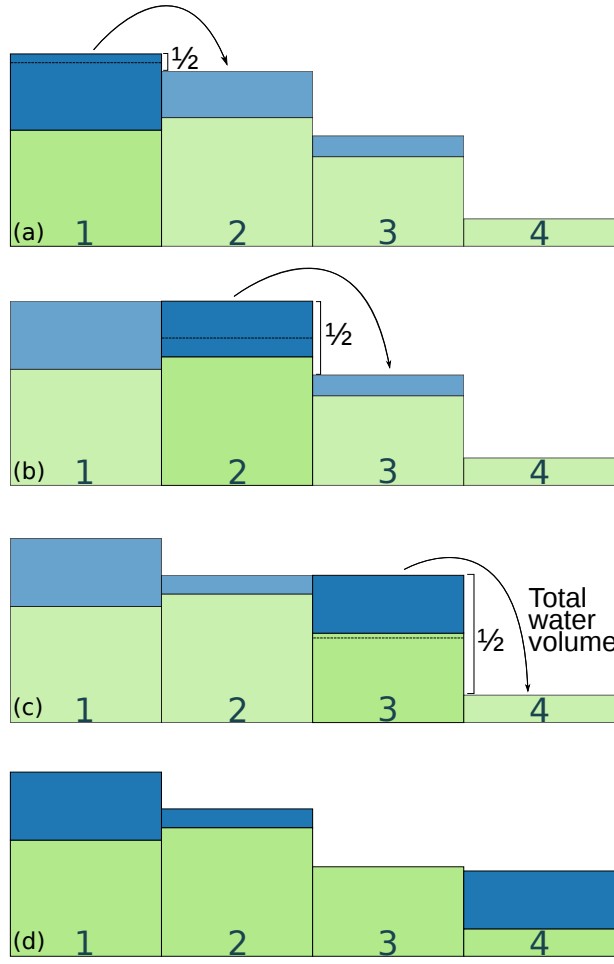

**Figure 3.** Water flow during a single iteration of FlowFill. In each iteration, water is moved starting with the highest (water + topography) cell and ending with the lowest. Each 'target cell' routes water into its steepest downslope neighbour. The amount of water moved from the higher cell to the lower one is the minimum of either all the water available in the target cell or half the difference in elevation between the target cell and its steepest downslope neighbour. This latter criterion ensures numerical stability, but slows convergence towards a solution. **(a)** Starting at the highest (topography + water depth) cell, half of the difference in topographic + water thickness height is moved from cell 1 to cell 2, this being the steepest downslope direction. **(b)** Cell 2 becomes the target cell. Half of the difference between cells 2 and 3 is moved to cell 3. **(c)** Cell 3 becomes the target cell. There is less water available in cell 3 than half the difference between cells 3 and 4, so all of the water from cell 3 is moved to cell 4. **(d)** Cell 4 becomes the target cell; if this were the edge of the domain, the water in cell 4 would flow out of the domain. A single iteration has been completed. Water will now start moving from cell 1 again, and the process will repeat until the solution converges.

elevations between the cells would have to be identical to several decimals. This process of moving water down the slope of the (topography + water) surface is repeated until a predefined criterion is met to indicate that the solution has converged: either a maximum number of iterations have been performed, or the maximum amount of water moved per iteration, $h_{max}$, has not changed by more than the user-defined threshold for 20,000 iterations ($i$) (Figure 4). This threshold defines the maximum value

of $|\Delta h_{max}|$ that will not reset the 'exit_counter' that terminates the FlowFill calculation (Figure 2), where:

$$|\Delta h_{max}| = |h_{max}(i) - h_{max}(i-1)| \tag{1}$$

Once the iterative downslope movement of water has been completed, FlowFill ensures that lake surfaces are flat. Due to the iterative algorithm in FlowFill, small spurious depressions can remain following convergence. To correct for these, we search for any pits (cells with no downslope neighbours) in the preliminary result. If a pit has one or more neighbours that contain

water, it should ultimately have received water from that neighbour had FlowFill been allowed to run for longer (which is computationally expensive), so we raise its water level to the level of the water-containing neighbour. Strictly speaking, this correction means that water mass has not been conserved. However, the change affected by this correction is small relative to the total water volume. The adjustment of the lake-level surface is thresholded to a maximum value at 1/10000 of the supplied runoff. The total volume of this adjustment ranged from 0 to 0.009% of the total water stored on the landscape at our study

sites.

Following this lake correction, FlowFill outputs three files. The first of two binary (32-bit floating point) files contains the flow-routing surface: topography with depressions filled or partially filled in accordance with the provided input runoff depth. The second contains only the depth of water that is lying on the landscape. The third file contains runtime messages in ASCII text format.

We implement FlowFill in MPI-enabled Fortran 90. This speeds calculations by splitting the domain into multiple horizontal bands with fringes that interact via the D8 flow-routing algorithm. Source code and compilation instructions are available at https://github.com/KCallaghan/FlowFill (Callaghan, 2019). Use is simplified through a provided text file for users to enter parameters, and a runfile. As an additional option, users can run FlowFill through a GRASS GIS extension, r.flowfill.

The gradual cell-to-cell water redistribution within FlowFill, along with its asymptotic approach towards equilibrium due

to its moving at most half of the head difference per iteration, can cause depressions to become 'overfilled' when the water-moving algorithm (Figures 3 and 2) terminates. This could happen when water has not been able to fully equilibrate over a depression, for example, when there is only a small path for water to escape a large area. These cases are not corrected inside FlowFill, but can be corrected in an additional step through comparison with the outputs of a flood-fill algorithm applied to the same initial DEM. Flood-fill algorithms fill all depressions fully to the level of their outlets, so we correct for overfilling

by (1) performing a flood fill using RichDEM's complete-depression-filling command (Barnes et al., 2014a, b; Barnes, 2016), and then (2) taking the minimum of the flood fill and the FlowFill outputs to produce the final result. This correction violates conservation of water volume, but the size of the adjustment is very small relative to the total volume of water stored on the landscape (See Section 4).

Flooded depressions have flat surfaces which can be problematic for flow routing, so flat areas were corrected using Rich-DEM to impose a gradient on these (Barnes et al., 2014a; Barnes, 2016). The result is a completed flow-routing surface that retains depressions based on conservation of water volume. In order to test FlowFill, we ran it with variable initial runoff depths on two landscapes. We then routed surface-water flow over the computed flow-routing surfaces and evaluated the degree of drainage integration at these locations.

## 4 Implementation

### 4.1 Example Data

We generated flow-routing surfaces using FlowFill from DEMs of two study regions. The first study region includes a reach of the Sangamon River in Illinois, USA, located at 39.97°N, 88.72°W. The low-relief plains left behind in this post-glacial landscape contain closed depressions that may impact hydrologic connectivity as a function of runoff (Lai and Anders, 2018). We resampled the 2.5 ft (0.76 m) resolution LiDAR DEM to 15 m resolution for our analysis. At several locations, bridges and other man-made structures cross the river channel at this study site. These are clearly visible on the lidar and artificially elevate the topography, creating blockages to water flow. These were manually removed using GRASS GIS before running FlowFill by digitising the problematic bridges, converting these to null cells, and then performing a bilinear interpolation to populate these cells with more realistic values. The second study region was the Río Toro basin, located mainly in Salta Province, Argentina, around 24.5°S, 65.8°W. Its steep topography was shaped primarily by fluvial processes, but has also been impacted by mountain glaciation, landslide dams, and tectonically-driven isolation of much of the basin from the foreland (though it has since re-incised) (Sobel et al., 2003; Tofelde et al., 2017; Trauth and Strecker, 1999). The DEM of this region was resampled to 120 m resolution from 12-m TanDEM-X data. The two landscapes differ in their topographic setting in terms of tectonics, glaciation, drainage integration, average slope, and spatial scale.

**Table 1.** Characteristics of the two study sites.

|  | Sangamon | Río Toro |
|---|---|---|
| Number of cells | 298200 | 638154 |
| Cell side length | 15 m | 120 m |
| High point | 224.413 m | 5970.257 m |
| Low point | 186.775 m | 1258.525 m |
| Average slope | 0.0239 | 0.301994 |
| Number of cells that are part of a depression | 20790 | 3128 |
| Depression volume per unit area | 0.01437 m | 0.05495 m |

In these examples, we prescribed uniform runoff across each DEM in order to easily compare depression filling and hydrologic connectivity (i.e. drainage integration). We tested runoff inputs of 1 mm to 15 m in the Río Toro basin and 1 mm to 20

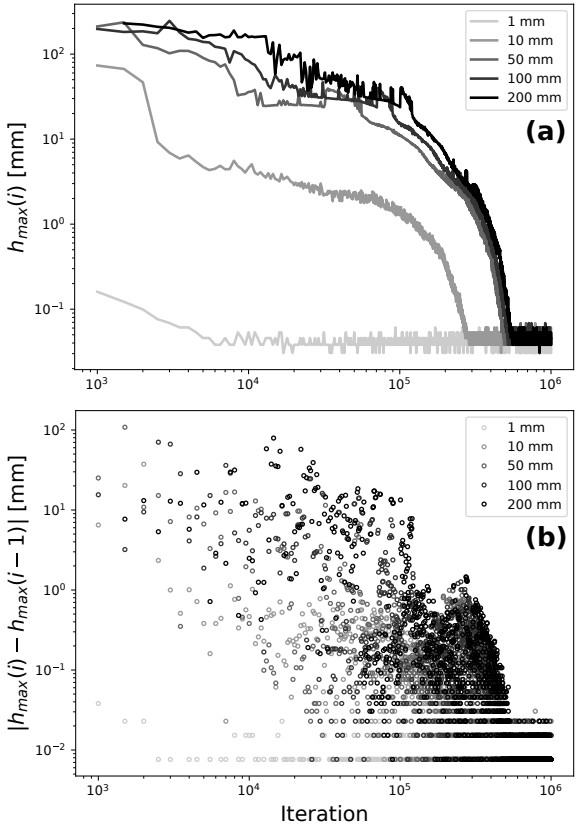

**Figure 4.** The maximum amount of water moving from one cell to another in a single iteration ($h_{max}$) is useful in deciding when to threshold FlowFill's result. Most water moves near the beginning of a model run. Panel **(a)** shows how $h_{max}$ changes for 5 different model runs (light grey to black, with depths of 1–200 mm given in the legend) at the Sangamon River site. Each run continues for 1,000,000 iterations ($i$) and each shows a distinct plateau at which $h_{max}$ ceases to consistently decrease. Prior to this plateau, we see some fluctuations in the amount of water moving per iteration as some water makes it to river channels and some depressions become filled, changing the evolving flow-routing surface. False plateaus represent periods of time in which the maximum amount of water moving per iteration does not significantly change. In order to avoid exiting the program early during one of these false plateaus, we conservatively wait for a plateau that lasts 20,000 iterations before thresholding our result. Panel **(b)** shows $|\Delta h_{max}|$ (Equation 1), the absolute value of the change in $h_{max}$ between the current and the previous iteration. Based on these data, we were able to select the threshold for this site as $|\Delta h_{max}| = 0.01$ mm. Once $|\Delta h_{max}| < 0.01$ for 20000 iterations, the model run saves the result and is complete.

cm in the Sangamon River basin (Table 2). For comparison, a typical storm event in the Río Toro basin drops ∼1 cm of rain (Castino et al., 2017, supplement), which equals the minimum runoff value discussed here. This minimum amount of runoff was not capable of filling many of the depressions in the landscape, and therefore our calculations indicate that some significant segmentation of the Río Toro catchment remains regardless of the size of the rainfall event. The median daily rainfall on a day

with rain near the Sangamon River is 3.3 mm, and the maximum recorded single-day rainfall is 99 cm (USGS 05590050: data from 01 October 2005 to 19 February 2019). This large range suggests that hydrologic connectivity in the Sangamon River basin depends on storm intensity.

**Table 2.** Runtimes for mass-conserving depression filling using FlowFill. Runtimes increase with the depth of applied runoff and on flatter landscapes (Table 1).

| Runoff depth [m] | Sangamon [min] | Río Toro [min] |
|:---:|:---:|:---:|
| 15 | – | 8.55 |
| 10 | – | 7.15 |
| 5 | – | 5.23 |
| 2 | – | 3.8 |
| 1 | – | 2.88 |
| 0.5 | – | 2.55 |
| 0.2 | 32.42 | 2.15 |
| 0.1 | 28.57 | 1.4 |
| 0.08 | 29.75 | – |
| 0.05 | 26.62 | – |
| 0.03 | 24.37 | – |
| 0.02 | 20.07 | – |
| 0.01 | 14.4 | 1.75 |
| 0.005 | 2.15 | – |
| 0.001 | 0.97 | 1.63 |

## 4.2 Results

We used FlowFill to fill depressions both at the Sangamon River reach (Figures 5 and 6) and in the Río Toro basin (Figures 7 and 8). We applied varying amounts of runoff to demonstrate differing levels of depression filling (Figure 9) and hydrologic connectivity (Figure 10). Both study sites contain persistent depressions that are unlikely to be permanently filled and connected via surface water, as well as smaller depressions that may be filled during modest rainfall–runoff events.

We varied the input runoff depth at the two sites, with a maximum value selected based on how much runoff was required to fill all depressions in each DEM, giving a result comparable to existing flood-fill algorithms. This maximum runoff depth was 0.2 m for the Sangamon River site (Figure 5), significantly lower than the 15 m runoff required for the Río Toro site (Figure 7). However, at the Río Toro site, most depressions were filled by significantly shallower runoff (0.1 m or less), and only a few large depressions persisted as we dramatically increased the initial runoff depth.

At both sites, deeper runoff fills more depressions, thus increasing hydrologic connectivity across the landscape (Figure 9). We define both drainage integration and hydrologic connectivity based on Strahler stream order (Figure 10 and Table 4), which

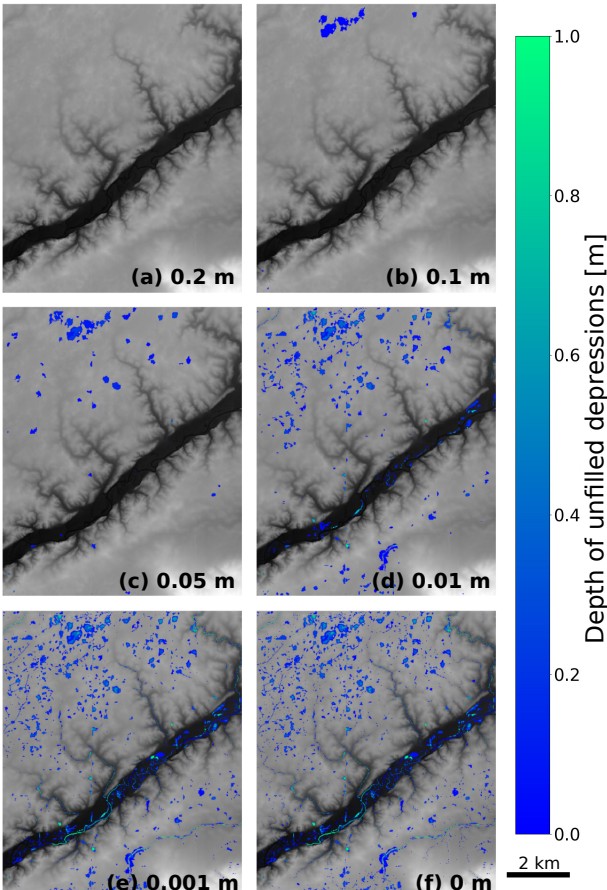

**Figure 5.** Depressions remaining with different amounts of starting runoff at the Sangamon River site. Deeper runoff fills more depressions. Unfilled depressions are shown for varying initial runoff depths: **(a)** 0.2 m, **(b)** 0.1 m, **(c)** 0.05 m, **(d)** 0.01 m, **(e)** 0.001 m, **(f)** 0 m (i.e. the input DEM with no changes made). DEM elevations are represented by a dark (low) to light (high) greyscale, while blue and green colours indicate the depths of depressions still present in the flow-routing surface. In the case of 0.001 m runoff, many depressions still remain, while with 0.1 m of starting runoff all but the largest depressions are filled. Depressions were fully filled with a starting runoff depth of 0.2 m.

we calculated using the 'r.stream.order' extension to GRASS GIS (Jasiewicz and Metz, 2011). The effect of variable runoff on drainage integration is more prominent in the Sangamon River landscape, due to its larger proportion of depressions that can, if unfilled, significantly break up the drainage network. The deep runoff required to completely fill all depressions (Figure 9) or produce higher-order drainage networks (Figure 10) implies that either heavy rainfall or a long period of rainfall – which
5  may or may not be plausible, depending on the climate and the hydraulic conductivity of the substrate – is necessary for the landscape to become fully hydrologically connected. Both landscapes contain a few larger depressions that persist once most other depressions have been filled, though based on their locations, they have somewhat less importance in setting overall

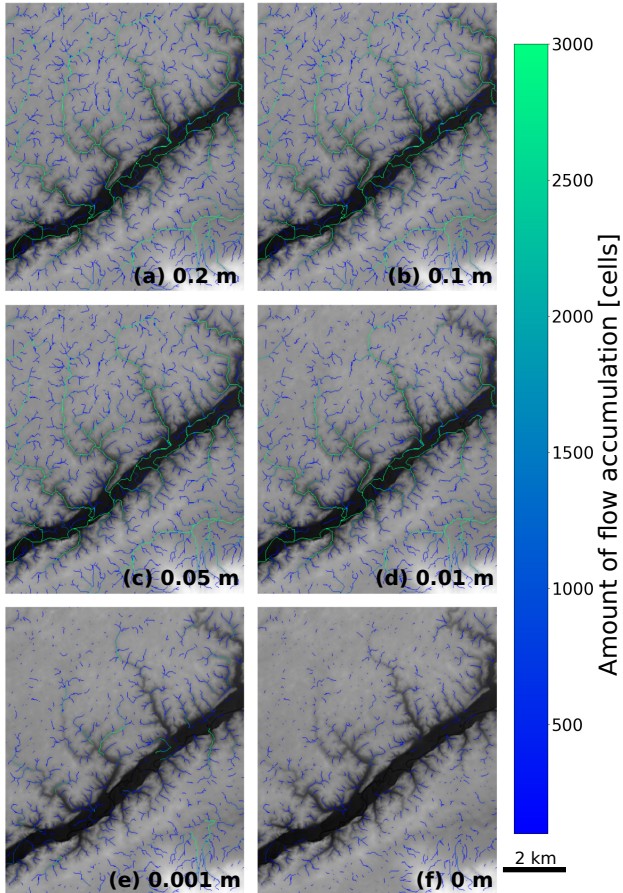

**Figure 6.** Drainage networks on partially-filled landscapes at the Sangamon River site. Flow networks were created using the FlowAccumulation method included in RichDEM (Barnes et al., 2014a; Barnes, 2016) with the result masked to show only cells with a flow accumulation greater than 100. Hydrologic connectivity changes depending on how many depressions remain in the landscape. The unfilled depressions associated with each panel are shown in Figure 5. The colours indicate the amount of flow accumulation in stream channels. Higher runoff depths applied to FlowFill fill more depressions and result in higher degrees of hydrologic connectivity. In panel **(a)**, with 0.2 m of runoff, all depressions were filled: drainage is fully integrated, and the result is identical to that for a flow-routing surface created using standard flood-fill techniques (e.g., Barnes et al., 2014a; Barnes, 2016). In panels **(b)**–**(e)**, decreasing amounts of starting runoff result in increasing segmentation of the stream network. Panel **(f)** shows the original DEM, which hosts only a few disconnected stream segments.

hydrologic connectivity, which saturates at runoff values below the maximum required to flood all depressions (Figure 10 and Table 4).

The processing time for FlowFill varies depending on the selected starting runoff depth, the number of cells in the domain, and the topographic structure of the site. Runtimes for our test cases varied from 0.97 minutes to 32.42 minutes (Table 2). We

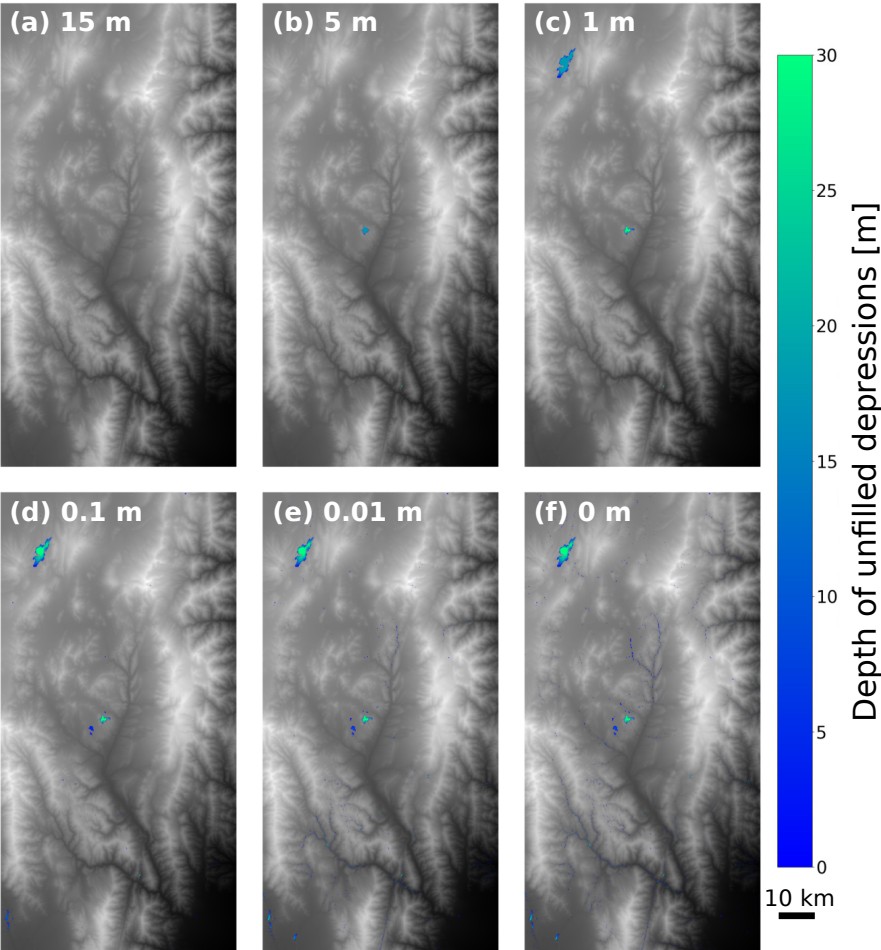

**Figure 7.** Depths of unfilled depressions in the Río Toro study area with starting runoff depths of **(a)** 15 m, **(b)** 5 m, **(c)** 1 m, **(d)** 0.1 m, **(e)** 0.01 m, and **(f)** 0 m (i.e. the input DEM with no changes made). DEM elevations are shown in greyscale, from dark (low) to light (high), while blue and green colours indicate the locations of depressions still present in the flow-routing surface. **(a)** In the case where 15 m of runoff was used, all depressions in the DEM were filled. **(b)** With 5 m of runoff, we see a persistent depression near the centre of the study region. **(c)** Another large depression appears at with 1 m of runoff, but most smaller depressions are filled. **(d,e)** With 0.1 m runoff and less, more depressions appear in the landscape. **(f)** All of the depressions appear on the original, unfilled DEM.

performed each calculation using 8 processors on an Intel i7-5820K CPU (3.30 GHz) on a desktop computer running Ubuntu Linux with 64 GB DDR3 RAM and a solid-state hard drive.

Due to the slight overfilling of some depressions in the outputs from FlowFill, a correction was performed, as discussed in Section 3. In the two sample study regions discussed in this paper, the volume of the adjustment for overfilling was insignificant,

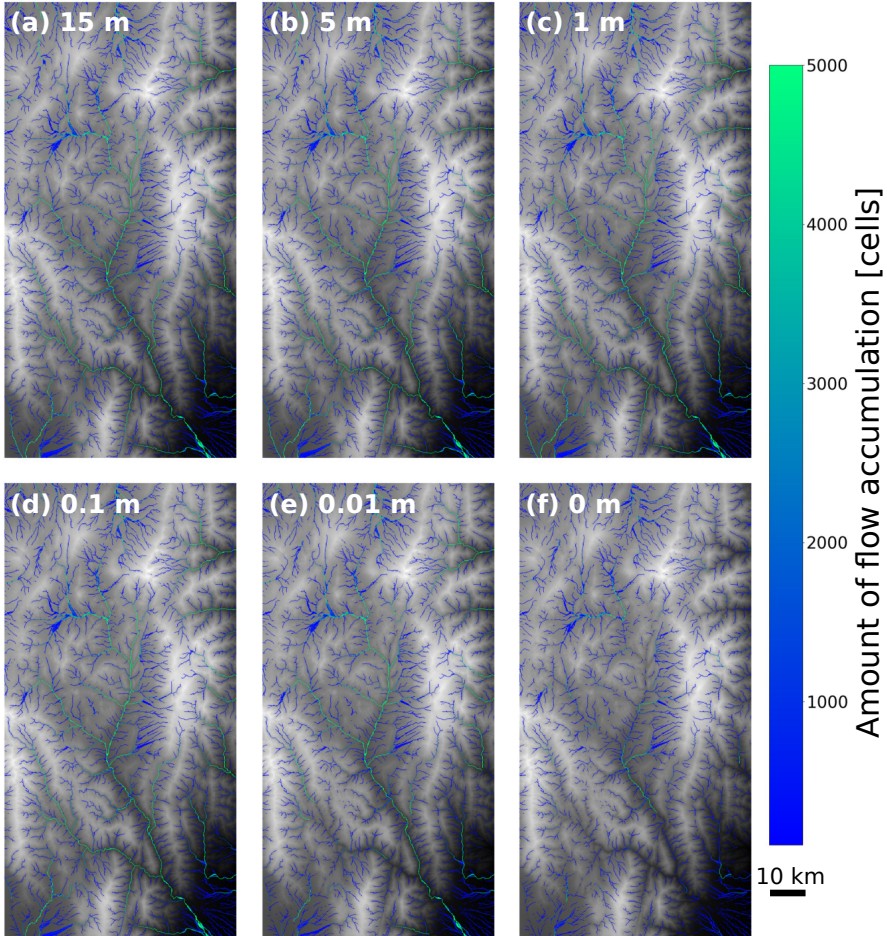

**Figure 8.** Drainage networks on partially-filled landscapes at the Río Toro site. Flow networks were created using the FlowAccumulation method included in RichDEM (Barnes et al., 2014a; Barnes, 2016) with the result masked to show only cells with a flow accumulation greater than 100. Hydrologic connectivity changes depending on how many depressions remain in the landscape. The unfilled depressions associated with each panel are shown in Figure 7. The colours indicate the amount of flow accumulation in stream channels. Higher runoff depths applied to FlowFill fill more depressions and increase hydrologic connectivity. **(a)** With 15 m of runoff, all depressions were filled so the result is identical to a flow-routing surface created with other flood fill techniques. The drainage is fully integrated. In panels **(b)** and **(c)**, with 5 m and 1 m of runoff depth respectively, hydrologic connectivity changes only slightly. The depressions that appear with these amounts of runoff are near the headwaters of the river network, making the changes in hydrologic connectivity in these cases minimal. In panels **(d)** and **(e)**, reduced runoff starts to create more disconnects in the stream network appear. Panel **(f)** shows the original DEM, which has the lowest degree of hydrologic connectivity. Strahler stream orders associated with each panel are shown in Table 4.

ranging from 0.003% to 0.29% of the total volume of water stored on the landscape. Cases in which the supplied initial runoff was deeper tended to have a slightly higher proportion of overfilling.

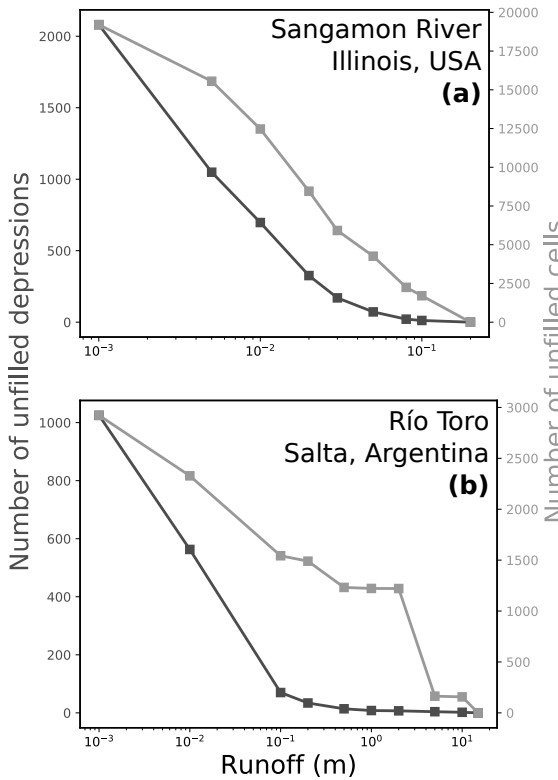

**Figure 9.** Number of cells (light grey) and depressions (dark grey) that remain unfilled under different starting runoff depths in **(a)** the Sangamon River basin and **(b)** the Río Toro basin sites. Similar trends are seen at both study sites, with higher runoff resulting in more depressions being completely filled. See supplementary information for exact figures.

In addition to these two study sites, we used FlowFill on a subset of the Sangamon study site at four different resolutions in order to assess how the resolution of the input data affects the results. The results of this analysis can be seen in Figures 12 and 13. A small subset of the Sangamon study site was selected in order to keep runtimes manageable at higher resolutions. The resolutions selected were 0.762 m (2.5 ft, the original resolution at which we obtained this data), 3 m, 5 m, and 15 m to match the resolution used for the entire Sangamon study site.

5     The results show that resampling data to a different resolution has an impact both on the number and morpology of depressions in the unfilled DEMs, and on the results obtained from FlowFill. When resampling to coarser resolutions at this site, the number of large depressions in the study area visually appears to increase, as seen in Figure 12. However, Table 3 shows us that the total number of depressions actually drastically decreases. This is due to the abundance of small depressions in the higher

10     resolution data. Instead, the total area of depressions present increases in lower-resolution data: the total area of depressions present in the unfilled DEM at 15 m resolution is almost 70% more than the area of depressions at 0.762 m resolution. This

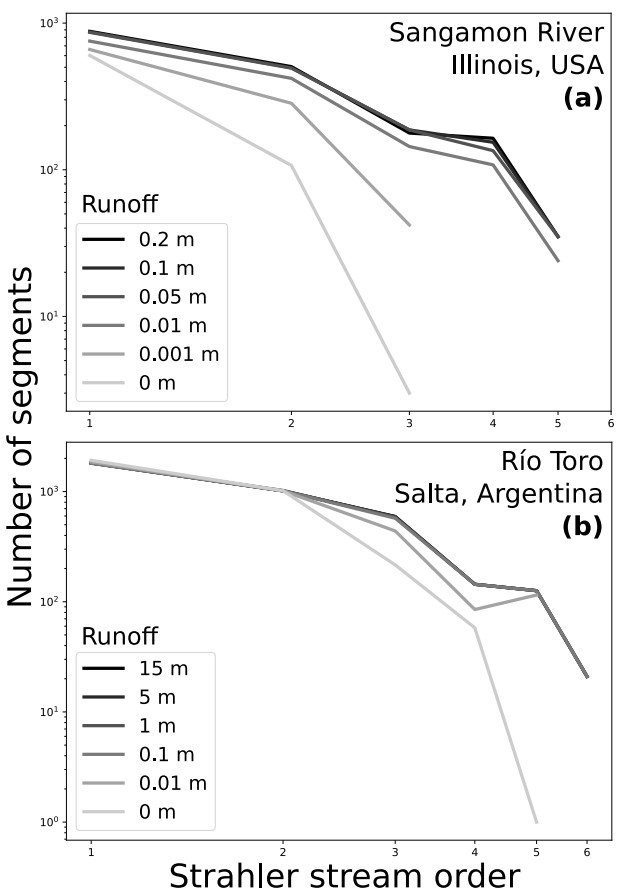

**Figure 10.** Strahler stream orders. As more water is added to the landscape, more depressions are flooded and drainage integration (and therefore hydrologic connectivity) increases. More stream segments overall exist on the landscape, and they become more connected, increasing the fraction of higher-order streams. Lines representing channel networks with shallower runoff depths overlie those representing deeper runoff for **(a)** the Sangamon River site and **(b)** the Río Toro site. We computed the stream orders in GRASS GIS (Neteler et al., 2012) using r.watershed (Metz et al., 2011) to compute stream networks, followed by r.stream.order to calculate the Strahler stream orders (Strahler, 1957).

trend is also reflected in the intermediate 3 m and 5 m resolutions. While these coarser resolutions have resulted in higher depression areas, the smoothing effect of resampling has also resulted in depressions becoming shallower, and hence, total depression volumes are smaller at coarser resolutions. This effect is less consistent: 15 m resolution exhibits the lowest total depression volume and 0.762 m has the highest, but the 5 m resolution DEM has a higher depression volume than the 3 m resolution DEM. Systematic changes in the shape of the landscape occur as resolution changes, but not all of these changes relate linearly to the change in resolution. All of these differences in depression number and morphology at different resolutions are an important reminder that the results of any landscape study based on remotely sensed data such as this are limited

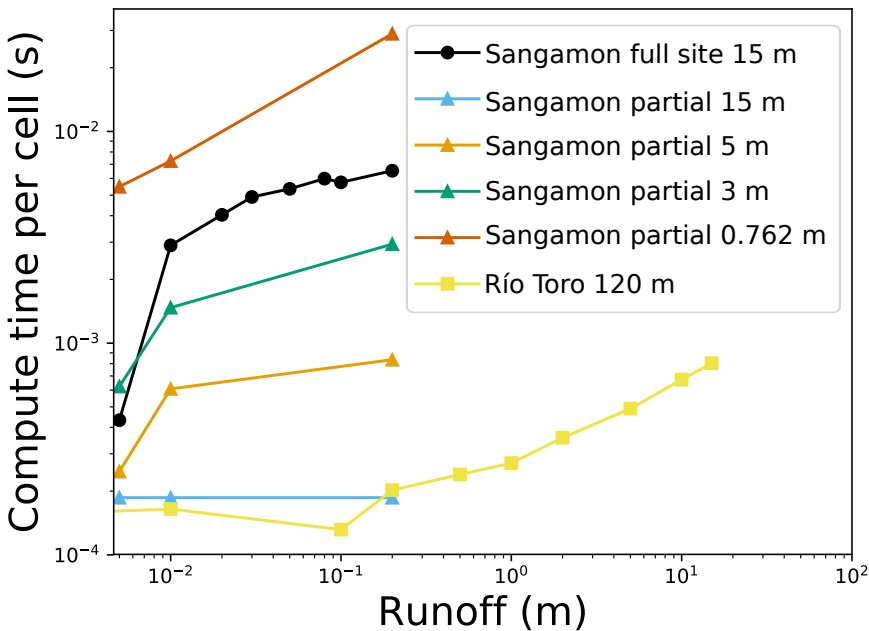

**Figure 11.** Time taken, in seconds per cell, for FlowFill to run to completion with different depths of starting runoff. Runs using larger starting runoff values take longer, and runs on a flatter spatial domain take longer. The Sangamon River DEM contained 298200 cells with total runtimes ranging from 59 to 1945 seconds. The Río Toro DEM contained 638154 cells with total runtimes ranging from 98 to 513 seconds. Details on cell counts and runtimes for the partial section of the Sangamon site are listed in 3.

by the accuracy of the input data, and any pre-processing steps may change this accuracy. The effects of different resolutions on depression storage are discussed in more detail by Abedini et al. (2006), and Dixon and Earls (2009) discuss the effects of different resolutions on watershed delineation and streamflow prediction.

We completed 12 runs of FlowFill, using DEMs with each the four resolutions, and with starting runoff depths of 0.2
5 m, 0.01 m, and 0.005 m (Figure 12 and 13, and Table 3). Regardless of the resolution, 0.2 m starting runoff filled all of the depressions, while lower amounts of runoff left some depressions unfilled. Overall patterns in depressions filled appear visually similar at all resolutions, with 15 m resolution showing the greatest difference from other resolutions. At 15 m resolution with 0.01 m runoff, several of the larger depressions on the southern edge have been filled, while these remained unfilled at finer resolutions. Drainage patterns also follow similar patterns at different resolutions, with the dominant river channels visible
10 at all four resolutions. Channel widths are inflated at coarser resolutions due to the larger cell size (Figure 13). Flow-routing pathways also differ between the 15 m resolution DEMs and those at finer resolution. When depressions are fully filled in the finer-resolution DEMs, a channel in the northeast corner of the DEM flows off the southern edge of the map. At 15 m resolution, however, the head of this channel is diverted, and flows off the eastern edge of the map.

Using data at the highest available resolution prevents loss or distortion of data and ensures that analyses are not introducing any additional errors due to downsampling. Unfortunately, runtimes for FlowFill for large data sets can become prohibitively long. This is a limitation of FlowFill, and more computationally efficient methods for dealing with depressions in flow-routing surfaces are needed. We begin to address this problem in a companion paper (Barnes et al., 2019).

5    Runtimes for this subset of the Sangamon site (Table 3) ranged from 2 seconds to 13 minutes for 15 m, 5 m, and 3 m resolutions with all input runoff depths, but escalated drastically to over 33 hours for 0.762 m resolution with 20 cm initial water depth. The reason for this nonlinearity lies in the fact that FlowFill moves water from cell to cell (Figure 3). Increasing the resolution both increases the total number of cells that must be calculated and requires more iterations of cell-to-cell water exchange for the water to move the same real-world distance. Runtimes times per cell are given in Figure 11.

**Table 3.** FlowFill runs on a subset of the Sangamon study site at four different resolutions and with varying amounts of starting runoff. Runtimes increase with the depth of applied runoff, and with increasing number of cells in the domain. Compute times per cell can be seen in Figure 11. The number of depressions is greater at higher resolutions: number of depressions scales linearly with the number of cells in the domain, where $number\_of\_depressions = 0.0034756 \times number\_of\_cells + C$. Depression areas tend to be greater at coarser resolutions.

| Resolution [m] | Number of cells | Runoff depth [m] | Runtime [min] | Number of un-filled depressions | Area of unfilled depressions [m$^2$] | Volume of unfilled depressions [m$^3$] |
|---|---|---|---|---|---|---|
| 15 | 10738 | 0.200 | 0.03333 | 0 | 0.00000 | 0.00000 |
| 15 | 10738 | 0.010 | 0.03333 | 16 | 207675.00000 | 36344.89860 |
| 15 | 10738 | 0.005 | 0.03333 | 31 | 256950.00000 | 42649.75318 |
| 15 | 10738 | 0.000 | – | 84 | 318150.000000 | 50542.00515 |
| 5 | 96996 | 0.200 | 1.35000 | 0 | 0.00000 | 0.00000 |
| 5 | 96996 | 0.010 | 0.98333 | 45 | 210525.00000 | 41696.87690 |
| 5 | 96996 | 0.005 | 0.40000 | 72 | 233275.00000 | 48746.75788 |
| 5 | 96996 | 0.000 | – | 414 | 283100.00000 | 56780.37223 |
| 3 | 269173 | 0.200 | 13.16667 | 0 | 0.00000 | 0.00000 |
| 3 | 269173 | 0.010 | 6.60000 | 112 | 201654.00000 | 40731.05525 |
| 3 | 269173 | 0.005 | 2.80000 | 218 | 216459.00000 | 47874.59637 |
| 3 | 269173 | 0.000 | – | 1254 | 245943.00000 | 55914.46614 |
| 0.762 | 4176000 | 0.200 | 2013.71667 | 0 | 0.00000 | 0.00000 |
| 0.762 | 4176000 | 0.010 | 504.56667 | 3112 | 172913.46062 | 47454.86143 |
| 0.762 | 4176000 | 0.005 | 381.55000 | 5091 | 177010.48468 | 54796.08098 |
| 0.762 | 4176000 | 0.000 | – | 14649 | 188244.20415 | 63816.53063 |

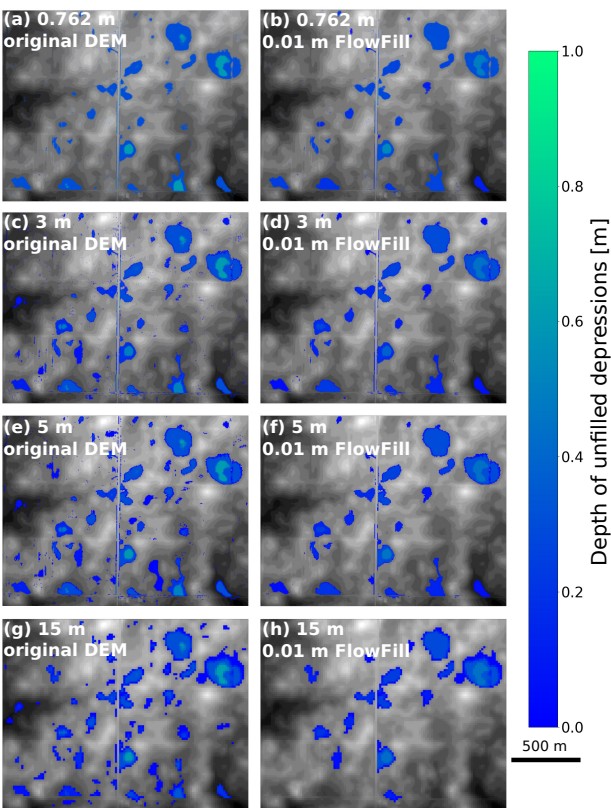

**Figure 12.** Depths of unfilled depressions in a subsection of the Sangamon study area at several different resolutions. The left column shows depressions existing in the unfilled DEM, and the right column shows remaining depressions after running FlowFill with 1 cm of starting runoff. The panels are **(a)** the unfilled DEM at 0.762 m (2.5 ft) resolution, **(b)** the 0.762 m resolution results after running FlowFill with 1 cm starting water, **(c)** 3 m resolution unfilled DEM, **(d)** 3 m resolution after running FlowFill, **(e)** 5 m resolution unfilled DEM, **(f)** 5 m resolution after running FlowFill, **(g)** 15 m resolution unfilled DEM, and **(h)** 15 m resolution after running FlowFill. DEM elevations are shown in greyscale, from dark (low) to light (high), while blue and green colours indicate the locations of depressions still present in the flow-routing surface. Resampling to coarser resolutions creates the visual impression of increasing the number of depressions since more large depressions are visible, but in reality the total number of depressions decreases as finer resolutions contain many small depressions, which are lost at coarser resolutions. The results after using FlowFill appear visually similar, with the exception of the 15 m resolution DEM, in which several larger depressions along the Southern margin were filled. Table 3 reveals that hundreds to thousands of less visible, smaller depressions were filled at finer resolutions.

## 5 Discussion

The flow-routing surfaces created by FlowFill account for water stored in the landscape and disconnects in the drainage network. The importance of such an approach is apparent because depressions persist in flow-routing surfaces even when the

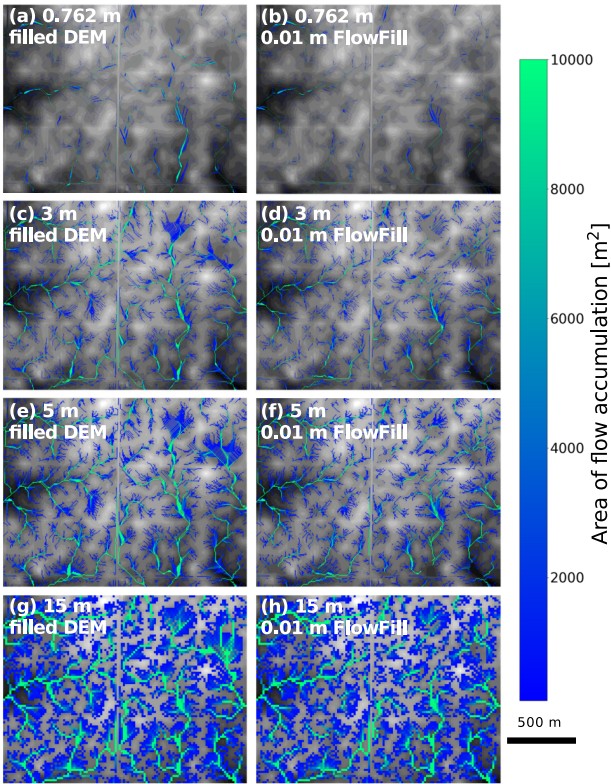

**Figure 13.** Drainage networks in a subsection of the Sangamon study area at several different resolutions. Flow networks were created using the FlowAccumulation method included in RichDEM (Barnes et al., 2014a; Barnes, 2016) with the result masked to show only cells with a flow accumulation greater than 500 $m^2$. The original, unfilled DEM supported very little drainage and is not pictured here. On the left are the fully-connected drainage networks occurring over a completely filled flow-routing surface. On the right are the partially connected networks resulting from the partially-filled surfaces created using FlowFill and 1 cm of starting runoff. The unfilled depressions associated with the right-hand column are shown in Figure 12. The colours indicate the amount of flow accumulation in stream channels, in $m^2$. The main river channels are consistently present at all resolutions.

prescribed initial runoff is deep. This indicates that preprocessing topographic data with algorithms that fill all depressions is likely to result in spuriously integrated drainage networks. We have demonstrated that this effect can occur in both high- and low-relief landscapes, and that in addition to correcting spurious depressions, true lake basins and swales must be taken into account. Spurious depressions can also be filled by runoff and will therefore also be corrected by FlowFill. Some of the available runoff will be used up in doing so. Because spurious depressions caused by data errors are likely to be small, these would be filled with even low amounts of runoff (Lindsay and Creed, 2006; O'Callaghan and Mark, 1984), though it is still possible that some depressions that remain unfilled are artefacts of data errors.

FlowFill provides users with a completed flow-routing surface, however, should a user prefer carving, breaching, or combined methods for depression removal, these can still be used in conjunction with FlowFill. The result obtained from FlowFill determines which depressions should be removed during creation of the flow-routing surface, and which should remain. Depressions that FlowFill has not completely filled can be masked out, while those which were completely filled can be selectively carved or breached. This allows a user to utilise their preferred depression removal method, while still being cognisant of the importance of retaining real-world depressions.

## 5.1 Hydrologic connectivity

We present FlowFill results at two locations with differing landscape characteristics (Table 1). The results include flow-routing surfaces with selectively filled depressions and the associated changes in drainage integration of the landscapes. A visual inspection of stream networks in cases where lower runoff values are used tells us that hydrologic connectivity is lower in these cases. This is quantitatively supported by the Strahler stream order data shown in Table 4. More higher-order streams occur when deeper runoff is used to create the flow-routing surfaces, hence filling more depressions. Channels were extracted using the FlowAccumulation functionality in RichDEM (Barnes et al., 2014a; Barnes, 2016) with a threshold of 100 units of accumulation.

The impact of using different amounts of runoff within FlowFill on the hydrologic connectivity of the landscape was more apparent in the Sangamon River basin study site, where more depressions were present. The presence of these depressions significantly reduced connectivity between stream segments. It is likely that FlowFill will be most useful in cases where the geologic and geomorphic history of a landscape produce a surface with many depressions, such as this post-glacial landscape. The high number of depressions seen may also be partially due to the finer resolution of this data relative to the Río Toro study site – density of depressions has been shown to relate to grid spacing as an inverse power law (Lindsay and Creed, 2005) – but this does not belie the finding that significant real depressions exist and impact hydrologic connectivity.

The ability to use FlowFill with varying user-selected starting runoff values makes it ideal for comparing network connectivity in wet versus dry seasons (Figure 10 and Table 4), or for analysing the effects of storms of different sizes. Shallow runoff inputs to FlowFill imitate real-world conditions with low amounts of rainfall (e.g. during the dry season). During these times, hydrologic connectivity is significantly reduced, and routing flow across a completely depression-filled landscape becomes unrealistic. Deep runoff inputs simulate wet seasons or flood conditions. Due to the associated greater hydrologic connectivity, more of the region contributes water to basin outlets.

## 5.2 Cellular-based modeling

FlowFill is a cellular automaton that models water flow across landscapes. While we designed FlowFill to fill depressions on digital elevation models in a way that conserves water mass, rather than to reproduce a physics-based transient flow response, we propose that its mechanism of moving flow between cells may be useful for modelling applications. The amount of water moved between cells at each iteration is gradient-based, meaning that FlowFill can approximate real transient flow in the landscape as a result of both topography (body forces) and water depth (pressure forces). Where the outputs of FlowFill

| | | | Río Toro | | | |
|---|---|---|---|---|---|---|
| Stream order | 15 m runoff | 5 m runoff | 1 m runoff | 0.1 m runoff | 0.01 m runoff | Original DEM |
| 1 | 1816 | 1816 | 1813 | 1805 | 1859 | 1922 |
| 2 | 1014 | 1014 | 1023 | 1017 | 1016 | 1018 |
| 3 | 593 | 593 | 584 | 573 | 439 | 216 |
| 4 | 144 | 144 | 144 | 144 | 85 | 58 |
| 5 | 126 | 126 | 126 | 126 | 115 | 1 |
| 6 | 21 | 21 | 21 | 21 | 0 | 0 |
| | | | Sangamon River | | | |
| Stream order | 0.2 m runoff | 0.1 m runoff | 0.05 m runoff | 0.01 m runoff | 0.001 m runoff | Original DEM |
| 1 | 879 | 871 | 865 | 754 | 661 | 602 |
| 2 | 505 | 496 | 493 | 420 | 284 | 107 |
| 3 | 178 | 187 | 187 | 144 | 42 | 3 |
| 4 | 164 | 154 | 135 | 108 | 0 | 0 |
| 5 | 35 | 35 | 35 | 24 | 0 | 0 |

**Table 4.** Number of streams of each Strahler order at each study site after flow-routing surfaces were created using different amounts of runoff. Deeper initial runoff was able to fill more depressions, integrating flow across them and building higher-order drainage networks.

diverges significantly from a true flow solution, for example, to the backwater equation, is that a parcel of water in FlowFill can move at most one cell per iteration, regardless of the underlying slope. Furthermore, FlowFill cannot accommodate different roughness values that would modulate flow velocity in one region versus another. With these limitations in mind, it could still provide a useful approach for simulations with approximately constant roughness and in which differences in elevation

5 are consistently less than the flow depth. Fortunately, such examples are common in geomorphology, and include reduced-complexity approaches towards simulating the dynamics of braided rivers (Murray and Paola, 1997) and river deltas (Liang et al., 2015b, a). With these uses in mind, users can view intermediate (pre-equilibrium) result outputs from FlowFill, after a set number of iterations or at frequent intervals. The compute time for an individual iteration of FlowFill ranged from $10^{-3}$ to $10^{-6}$ s for the regions discussed in this article.

10 **5.3 Limitations**

While we have created a way to handle the problem of real-world depressions in a DEM, FlowFill does have some limitations. Firstly, FlowFill requires significantly more compute time than flood-fill methods that fully fill DEM sinks (Barnes et al., 2014a; Barnes, 2016; Schwanghart and Scherler, 2014). Secondly, it creates flat areas in DEMs, which present their own challenges for flow routing. Thirdly, the threshold value for $|\Delta h_{max}|$ is distinct in different landscapes, and as such is a user-defined

parameter. Finally, if an input topography contains three-dimensional structures such as bridges, FlowFill can cause water to artificially dam behind them.

FlowFill can be time-consuming to run, especially for large DEMs, high starting runoff depths, or relatively flat study sites. Runtimes for each of the results shown here are shown in Table 2, with the compute time per cell shown graphically in Figure 11. This may make it an unappealing choice in cases with large study areas or very high-resolution data. Therefore, while FlowFill can route runoff to create flow-routing surfaces, a more computationally efficient solution to the problem outlined in this paper will permit faster analyses of a wider range of DEMs.

Like some other depression-filling algorithms, FlowFill produces flat areas where it fills depressions. Post-processing these into a gentle slope may be required in order to create reasonable or visually appealling flow networks. Fortunately, tools to do so efficiently already exist. To produce our drainage networks for the stream-order calculations (Figure 10), we used RichDEM to impose a gradient on flat areas (Barnes et al., 2014a; Barnes, 2016) as the final step in constructing each flow-routing surface.

While a single criterion for convergence on a final flow-routing surface would be ideal, we were unable to find one, and instead have left this as a user-selected parameter. We attempted to use the maximum amount of water moved between two cells in an iteration, the total amount of water moved in an iteration, the rate of change in the maximum amount of water moved between two cells in an iteration (averaged over various time windows), and the root-mean-square error of the linear regression that created the aforementioned slope. We also attempted each of these methods while normalizing for the initial amount of applied runoff. None of these approaches were able to collapse the response curves of flow over the landscape. However, we do observe that the maximum amount of flow between two cells in a single iteration asymptotes to a consistent value in each landscape. We therefore selected the exit criteria based on a time when the change in the maximum amount of flow between cells from one iteration to the next is below some small, user-selected threshold. It is necessary for a user to select this threshold value since the amount of noise after the plateau has been reached varies from one landscape to the next. As a result, we suggest that users who want to test multiple initial runoff depths first run FlowFill with a modest amount of runoff (to speed compute time: Figure 11) in order to create a maximum water depth moved vs. iteration curve like those shown in Figure 4. From this, users may pick a threshold value for $|\Delta h_{max}|$. Selecting a value that is too large will cause FlowFill to exit before reaching this plateau, while a value that is too small will cause FlowFill to continue running for its maximum limit of 1,000,000 iterations, resulting in a long compute time. Based on our two study areas, suitable threshold amounts are landscape-specific, but appear to be agnostic to the amount of runoff selected for a given landscape. Following this approach, we chose thresholds of 0.01 mm (Sangamon) and 1 mm (Río Toro).

All depression-filling algorithms can produce 'lakes' as artefacts where the two-dimensional topography does not represent efficient three-dimensional flow paths – such as flow under bridges or through culverts (Lindsay and Dhun, 2015; Passalacqua et al., 2012). FlowFill is especially sensitive to these, as their damming effect can also create bottlenecks that significantly increase the number of iterations required to evacuate the water behind them, even when a narrow flow path exists to bypass them. Even after convergence, these areas often require the additional step to reduce overfilling, discussed in Section 3. This common problem further motivates work to remove these artificial blockages from rivers in DEMs for flow routing (Abdullah et al., 2012).

# 6 Conclusions

Common and efficient downslope flow-routing algorithms must be run across surfaces that properly represent a true surface-water-potential surface. As modern DEM resolution and accuracy increases, this requires that DEM depressions be appropriately filled. We have developed an algorithm, FlowFill, that fills only those depressions on a landscape which would become filled under reasonable runoff conditions. This allows for the existence of real depressions and hydrologic disconnects in the landscape.

By adding more realistic surface-water hydrology to flow routing, FlowFill's ability goes beyond that of static flood-fill algorithms and enables scientists to examine dynamic hydrologic connectivity. While FlowFill effectively solves this important problem, its long runtimes for larger datasets can make its use inconvenient. Future advances towards a more computationally efficient methodology will aid in the longer-term goal of linking real-world data with algorithms that harness the emerging power of far-reaching and high-resolution topographic data.

*Code availability.* The latest version of FlowFill is available at https://github.com/KCallaghan/FlowFill. The most recent version at the time of publication, 1.1.0-beta (Callaghan, 2019), is archived at https://doi.org/10.5281/zenodo.2586244. The GRASS extension r.flowfill is available at https://github.com/OSGeo/grass-addons/tree/master/grass7/raster/r.flowfill.

*Author contributions.* KLC developed the algorithm behind FlowFill, wrote its source code, ran the tests, and was the primary author of the manuscript. ADW provided guidance in the parallelization, suggested the test sites, and co-wrote the manuscript.

*Competing interests.* The authors declare that they have no conflict of interest.

*Acknowledgements.* The Deutsches Zentrum für Luft- und Raumfahrt (DLR) provided 12-meter TanDEM-X DEM coverage of the Río Toro catchment via proposal DEM_GEOL1915 awarded to Taylor Schildgen, Andrew Wickert, Stefanie Tofelde, and Mitch D'Arcy. The authors acknowledge the Minnesota Supercomputing Institute (MSI) at the University of Minnesota (http://www.msi.umn.edu) for providing resources that contributed to the research results reported within this paper. Jingtao Lai and Alison Anders provided a copy of their Sangamon River DEM. KLC was supported by the University of Minnesota Department of Earth Sciences HE Wright Footsteps Award and Junior F Hayden Fellowship, and by start-up funds awarded to A. Wickert by the University of Minnesota.

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
