# Peer review of "Computing water flow through complex landscapes, Part 1: Incorporating depressions in flow routing using FlowFill"

_Earth Surface Dynamics, 2019_

## Referee Comment (RC1) · Stuart Grieve (Referee) · 1 Apr 2019

Firstly I would like to thank the authors for this contribution, in my opinion this is an issue which has urgently required attention for some time and so I am pleased to see it is being worked on. This manuscript presents a new method for the hydrologic correction of DEMs, based on the distribution of hydrologically meaningful surface runoff across the landscape. This new algorithm allows for the preservation of pits or depressions in the landscape which the authors argue are likely to be natural features, rather than processing artifacts as often assumed. In doing this, the amount

of hydrologic connectivity of a landscape can be estimated for a range of runoff regimes. The effectiveness of this algorithm is demonstrated using DEMs of two distinct landscapes, both known to contain natural depressions. This manuscript and the algorithm it presents should be of broad interest to the surface process community as it begins to integrate hydrologically meaningful information to a previously abstract process.

**General comments**

Reading through this work, I often found that my questions were answered in a later section of the manuscript, and the inclusion of the Limitations section has neatly outlined many of the issues I would otherwise have raised. The first issue I have with this manuscript is the two test DEMs which have been selected. A prime motivation for this work, one that is discussed in detail in the introduction, is that as we move from lower resolution DEM products (e.g. SRTM, ASTER) to higher resolution LiDAR derived products, the depressions we identify and traditionally fill are more likely to be natural features. So it was disappointing to see the two datasets being employed having a grid cell size of 15 and 120 meters. It is unclear as to why this algorithm was not tested on higher resolution data, if there is a valid reason for this it should be communicated to the reader, otherwise, I suggest using some higher resolution data for the evaluation which is more typical of the work being done with high resolution topographic datasets in our community.

Following from this, there is no detail provided on the resampling method employed for either of the datasets. Depending on the technique used this could introduce noise into the topographic data, which may cause bias in the results by either creating new depressions or removing existing ones. I am assuming that the original data was also

in gridded form, and so this effectively means that the data you are using has been resampled twice. I would like to see a clear justification of this double resampling and also some confirmation that it id not biasing the results.

It is not clear to me how easy it will be to use the output of this code in existing topographic analysis packages. If we do not use a maximum runoff depth, some depressions will be preserved in the data, which will cause many popular flow routing algorithms to fail (eg Braun and Willett, 2013). If I am correct in this assumption, some more detail in the manuscript drawing out this distinction would be helpful. I think the power of this technique comes not as a replacement to existing hydrologic correction, but as a tool to identify connectivity under different runoff regimes. For example, would it be possible to couple this code with a rainfall generator such as STORM (Singer et al., 2018) to develop better understanding of landscape connectivity and surface runoff under variable rainfall conditions?

There is no discussion of the suitability of a D8 flow routing scheme as a representation of real runoff behavior. Doing a full analysis of how differing schemes (D-infinity, MFD) may impact the results would be well beyond the scope of this contribution, but some discussion around these issues would be a valuable addition. I think this is particularly important as the basis of this algorithm is that it is physically-based and so it is important to justify the abstraction of using D8 rather than a more "correct" routing scheme.

**Line by line comments**

In addition to the issues mentioned above, I have some more general minor line by line comments:

Page 1, Line 13 - The discussion of time here is subjective, and dependent on dataset size, compute speed and landscape complexity. I would reframe this sentence around efficiency rather than time.

Page 2, Line 5 - Suggest changing 'of the landscape, the climate' to 'of landscapes, climates'.

Page 2, Line 7 - Replace 'equations' with 'equation'.

Page 2, Line 9 - Cite Braun and Willett (2013) here as well.

Page 3, Line 13 - Fix brackets around citation at end of line.

Page 4, Line 15 - Figure 3 is referenced before figure 2 here.

Page 6, Line 3 - What about integer DEM datasets? In that case ties could be very common, is there a better way to address this?

Page 14, Line 3 - I am asking a lot here, but is there any way to differentiate between true and spurious depressions?

**Figures and Tables**

Figure 2 - Is it possible to add labels to the cells? In the caption you refer to cells by their number and it would be nice to see those in the figure.

Figure 5, 6, 8 - The captions do not mention which dataset is being shown, its clear from the text, but would be helpful when skimming the paper to have this information to hand.

Figure 10 - I don't think you explain anywhere how you have extracted your channels, it looks like you have set an area threshold, and as long as this is constant within each dataset this won't matter much but it would be nice to see it stated explicitly.

**Code**

I am happy that the algorithm being described in the paper is what is being implemented in the cited github repo, and would like to commend the authors for the provision of open source, licensed code with detailed documentation.

– Stuart Grieve

**References**

Braun, J., Willett, S. D. (2013). A very efficient O (n), implicit and parallel method to solve the stream power equation governing fluvial incision and landscape evolution. Geomorphology, 180, 170-179.

Singer, M. B., Michaelides, K., Hobley, D. E. (2018). STORM 1.0: a simple, flexible, and parsimonious stochastic rainfall generator for simulating climate and climate change. Geoscientific Model Development, 11(9), 3713-3726.

---

## Referee Comment (RC2) · Wolfgang Schwanghart (Referee) · 8 Apr 2019

In this paper, Kerry Callaghan and Andy Wickert describe a method that computes water flow through complex landscapes represented by digital elevation models (DEMs). The manuscript mainly deals with the description of the algorithm and its testing with two moderately sized DEMs. As the title indicates, the manuscript is part of a two- or more-part manuscript which suggests that there will be an application or further testing of the FlowFill algorithm in another manuscript. At the time of writing this review, the second part was not yet available.

[Figure]

Major comments

1. The authors set their work in the context of DEM preprocessing. They review a number of preprocessing techniques such as filling and carving and the hybrid methods that exist and conclude that these methods fail to resolve the problem of hydrological connectivity. Filling (or carving) all depressions would not reflect the actual water flows and the existance of internal drainages whose occurrence is highly dependent on the magnitude of runoff events. I totally agree. Blindly applying existing DEM preprocessing algorithms can result in flow paths that do not reflect actual flow patterns. However, is FlowFill a viable solution to the problem of deciding which topographic depressions should be filled (or carved) or not? I think that FlowFill is quite an elegant solution to the problem. Routing water downstream and filling topographic depressions until they spill over (thus sequentially filling nested pits which some algorithms struggle with) reflects the actual water movements and puts each sink in relationship with its upstream area. The drawback of this approach, however, is that the filling of a particular sink is highly dependent on what happens upstream. An insignificant sink along a river may not be filled, because there are sinks upstream that hold water back. Sink removal is thus highly dependent on the topology of the network of topographic sinks. Of course, this may be an issue if sinks are rather a data artefact than true sinks. But DEM preprocessing techniques actually deal with and correct for these artefacts. If FlowFill is not designed for this, and line 1 on page 18 actually suggests this, than FlowFill is not really a DEM preprocessing technique. Rather, it is a highly simplified simulation tool that models water flow across landscapes without loss through infiltration or evaporation.

2. The flow routing model is a highly simplified representation of flow across complex terrain. Still, it is computationally intense because it takes time to converge. The DEMs that the authors test are very small compared to globally available DEMs and commonly available LiDAR DEMs (Sangamon has 550x550 cells (provided that the DEM is square), Rio Toro has 800x800 cells). I wonder whether the objective of determining the amount of water in each topographic sink given a runoff volume can be obtained

more easily. My idea draws from what I have stated above: modelling water flow using a network of sinks. Specifically, this could be done by following steps of a bucket model:

- Filling the DEM

- Calculating the volume of each sink

- Deriving flow directions

- Deriving drainage basins of each sink. The drainage basin of each sink thereby should exclude the basins of upstream sinks.

- Compare sink volume to total runoff in the upstream drainage basin.

- Topologically order sinks from top to bottom.

- Route excess water to downstream sinks if runoff volume exceeds sink volume.

This procedure would not explicitly model water flow across each sink. But it would help decide which sinks spill over and which not. Clearly, the proposed algorithm becomes increasingly complex if nested pits exist. But that is something one could deal with in a network of sinks.

Overall, I like the paper. However, I think that the paper would benefit from placing more emphasis of FlowFill being a modelling tool to study hydrological connectivity of complex landscapes, rather than a preprocessing technique. I encourage the authors to reconsider their algorithmic approach to this problem because working with network of sinks might be much faster.

Minor comments:

**3-9f** This should not come as a self-promotion of my own work, but quantile carving (Schwanghart and Scherler 2017) may also be a technique worth mentioning in the context of hybrid DEM preprocessing techniques.

**4-15** Figure 2 was not mentioned before. Be consistent with the order of figures and the order of how their references appear in the text.

**8-13** These numbers are not consistent with Table 2 which shows that Rio Toro was also tested with 1 mm runoff.

**13-2** "more realistic"? More realistic than what? Where do you demonstrate this statement? Validation is lacking. Will this be handled in the second part of the manuscript?

**Figure 11** The y-axis is difficult to read because there is only one tick label.

**20-6** You might want to mention the project ID of the TanDEM-X DEM project.

References:

Schwanghart, W. and Scherler, D.: Bumps in river profiles: uncertainty assessment and smoothing using quantile regression techniques, Earth Surface Dynamics, 5(4), 821–839, doi:10.5194/esurf-5-821-2017, 2017.

---

## Author Comment (AC1) · 23 May 2019

We thank the two referees, Dr Stuart Grieve and Dr Wolfgang Schwanghart, for their constructive comments on our work.

I will address firstly the comments from the first referee, Dr Grieve.

**Comment**: It is unclear as to why this algorithm was not tested on higher resolution data, if there is a valid reason for this it should be communicated to the reader,

otherwise, I suggest using some higher resolution data for the evaluation which is more typical of the work being done with high resolution topographic datasets in our community.

**Response**: We are happy to include some results and a brief discussion of the use of this algorithm on higher-resolution (2.5 ft) data in the final submission. We will provide an example using a subset of the Sangamon River site at this higher resolution. It does, of course, make sense to do so in light of the data that are available today and our specific discussion of the importance of distinguishing real depressions using high-resolution data. There were two reasons that we did not include this in our initial submission:

1) We are very aware of the limitations of our method in terms of computational efficiency. Processing times for high-resolution data will be long and while we think that the motivations for this work are particularly important where high quality, high resolution data are available, we also recognise that a more efficient method for dealing with the problem is needed in such cases. However, FlowFill does provide an intuitive way to visualise and map flow routing and ponding across real landscapes. The importance of highlighting the issue of real-world depressions outweighed the computational difficulties in our own method of dealing with it.

2) I decided it was more important to highlight the applicability of this method in morphologically different terrains – i.e. high versus low average slope, many versus few natural depressions – than to focus on a smaller, higher resolution study area. However, there is no reason not to include the high resolution data in our final submission.

**Comment**: There is no detail provided on the resampling method employed for either of the datasets. Depending on the technique used this could introduce noise into
the topographic data, which may cause bias in the results by either creating new depressions or removing existing ones. I am assuming that the original data was also in gridded form, and so this effectively means that the data you are using has been resampled twice. I would like to see a clear justification of this double resampling and also some confirmation that it is not biasing the results.

**Response**: We resampled each DEM by taking the mean value of the high-resolution cells within each low-resolution, larger cell. The reason for doing so was to produce sample study DEMs that were appropriate for our purpose of providing examples of the type of results that FlowFill can produce, in an amount of time that we found acceptable using the hardware at our disposal.

Resampling of data may cause bias in the results, and this is something that has been studied by other authors (e.g. Dixon and Earls, 2009). This is an issue at any time when resampling is performed, for any DEM processing method, flow accumulation, or hydrologic analysis performed. Coarsening the DEM generally results in a decrease in the total number of depressions present, since many of the smallest depressions are eliminated (MacMillan et al, 2011). Since our intention here is to provide a proof of concept, rather than to specifically guide work at the locations we have studied, any bias that may have been introduced was not detrimental to our results.

While I have not yet tested this, it seems likely that using FlowFill for the same study area at different spatial resolutions will produce slightly different results. As mentioned above, we will include an example using a portion of the Sangamon River site at a higher resolution in the final submission; we will also include a comparison between results obtained at the two resolutions used. The selection of an appropriate resolution and whether or not resampling is acceptable will depend upon the user's needs and discretion.

[Figure]

**Comment**: It is not clear to me how easy it will be to use the output of this code in existing topographic analysis packages. If we do not use a maximum runoff depth, some depressions will be preserved in the data, which will cause many popular flow routing algorithms to fail (eg Braun and Willett, 2013). If I am correct in this assumption, some more detail in the manuscript drawing out this distinction would be helpful. I think the power of this technique comes not as a replacement to existing hydrologic correction, but as a tool to identify connectivity under different runoff regimes. For example, would it be possible to couple this code with a rainfall generator such as STORM (Singer et al., 2018) to develop better understanding of landscape connectivity and surface runoff under variable rainfall conditions?

**Response**: It is true that some flow-routing algorithms will fail once they detect depressions still in the data. However, this behavior is algorithmic instead of realistic, and a main point of our work is to demonstrate the importance of considering natural depressions in the landscape along with times in which these will (or will not) overflow.

The referee's comment about the difference between being a replacement for existing hydrologic corrections or a tool to identify connectivity in fact identifies both sides of the same question. Existing flow-routing algorithms include an assumption of integrated drainage that may or may not be true. Our goal here is to take a step towards bridging that gap.

To directly answer the question about pre-processing: a user would have to select a flow-routing method that allows depressions to be present. For example, when creating figures for this paper, I (Callaghan) used the FlowAccumulation method from RichDEM, which worked as expected. A GRASS GIS user could use r.watershed, which allows inclusion of a layer showing which depressions on the landscape are real. For our revised manuscript, we will create a GRASS extension for FlowFill, which would make it more accessible to users.

With reference to coupling with a rainfall generator, the simplest way to use these together would be to use generated rainfall as the 'runoff' input into FlowFill. It is in principle possible to use FlowFill with variable runoff inputs, and this functionality will become available in the next release of FlowFill. A user will be able to use an input constructed from any model or real-world data they prefer, or they can elect to use the single-value runoff we have used in our examples.

**Comment**: There is no discussion of the suitability of a D8 flow routing scheme as a representation of real runoff behavior. Doing a full analysis of how differing schemes (D-infinity, MFD may impact the results would be well beyond the scope of this contribution, but some discussion around these issues would be a valuable addition. I think this is particularly important as the basis of this algorithm is that it is physically-based and so it is important to justify the abstraction of using D8 rather than a more "correct" routing scheme.

**Response**: The conceptual basis of our method – moving water downslope from cell to cell in such a way that the maximum amount of water moved makes the two cells equal in total elevation – requires a D8 or D4 environment. Using something like D-infinity, which may move water to more than one cell at a single step, would require some reconceptualization of the technique. Any approach using D-infinity or MFD would have to be significantly different than the one used in FlowFill.

**Comment**: Page 6, Line 3 - What about integer DEM datasets? In that case ties could be very common, is there a better way to address this?

**Response**: While we believe that it would in general be best to avoid integer datasets in cases where more precise data is available, some users may want to use integer

datasets. Even then, ties should still be less common than one may expect, since the algorithm includes the height of the topography plus water thickness when selecting a direction. If a constant amount of water is applied across the landscape, ties would be more common at first, but they should be significantly reduced after the first few iterations. Given the possibility that ties may present a problem in some cases, the next release of FlowFill will include an option to treat ties by selecting a random direction rather than a preferential direction. Since the use of randomness will make the results non-deterministic, the user will choose which method they prefer at runtime.

**Comment**: Page 14, Line 3 - I am asking a lot here, but is there any way to differentiate between true and spurious depressions?

**Response**: Unfortunately, that is asking a lot, which is a big reason why this problem still exists. It appears that the best way to do so is by ground inspection (Lindsay & Creed, 2006) which is often impossible. Other methods, such as modelling approaches, require assumptions about what makes a depression spurious (for example, small size). If we do assume that spurious depressions are small in size (few grouped cells having erroneous data, or depressions being smaller than the error margin of the data), then FlowFill should fill these with even small amounts of runoff.

In some cases, spurious depressions may be large, for example when a ravine is dammed by a single spuriously high cell that creates an artificial depression behind it. FlowFill will not deal well with such cases, but nor will other flood-fill methods. In such a case, a breaching or carving algorithm would do best, but again would ignore true depressions. I recommend a combination of FlowFill and a knowledge-based approach to correcting bad data. For example, when preparing my own data for these examples, I noticed that a man-made bridge in the Sangamon site caused water to dam behind it. Since in reality water would flow under the bridge, I manually lowered these cells by cropping them out and extrapolating data across the small gap.
Smaller specific line and figure comments have been dealt with as suggested, and we thank Dr. Grieve for his diligence.

Comments from the second referee, Dr Schwanghart:

**Comment**: The drawback of this [Flowfill] approach, however, is that the filling of a particular sink is highly dependent on what happens upstream. An insignificant sink along a river may not be filled, because there are sinks upstream that hold water back. Sink removal is thus highly dependent on the topology of the network of topographic sinks. Of course, this may be an issue if sinks are rather a data artefact than true sinks. But DEM preprocessing techniques actually deal with and correct for these artefacts. If FlowFill is not designed for this, and line 1 on page 18 actually suggests this, than FlowFill is not really a DEM preprocessing technique. Rather, it is a highly simplified simulation tool that models water flow across landscapes without loss through infiltration or evaporation.

**Response**: I agree in part with this comment: FlowFill is a simplified simulation tool that models water flow across landscapes. It has several potential uses, one of which is filling depressions in a landscape under a certain amount of runoff, and is the focus of this paper. I also agree that using this method, sink removal is highly dependent on network topology and that issues may arise in cases where sinks are data artefacts. This is an issue inherent to the method: by assuming that all depressions in the DEM are true landscape features, we ignore cases where depressions really are data errors. Even if spurious depressions are filled by FlowFill, they have 'used up' some of the water available on the landscape and this water will not be available further downstream.

This is a limitation of FlowFill, but does not negate its usefulness for this application.

First of all, spurious depressions are likely to be filled with even low runoff values

since they are often a result of small uncertainties (Lindsay & Creed, 2006). Secondly, differentiating real depressions from erroneous ones is challenging. Lindsay and Creed (2006) discuss a few ways of doing so. The most reliable method is ground inspection, but this is often impractical. Other methods, e.g. modelling, require assumptions to be made about what constitutes a real depression. These methods may also be flagging a certain depression as real when it is not, or vice versa. However, it is important to note here that there is nothing stopping a user from first using a different method to remove depressions that they believe to be spurious, and then using FlowFill to see how hydrologic connectivity may change under different amounts of starting runoff. A user may also choose to use the output of FlowFill in more complex ways than we have here: for example, if they prefer carving to remove depressions, they may run FlowFill to determine which of the depressions should be removed from their study area, and then use a carving algorithm that allows them to preserve they depressions that they wish to keep (those that FlowFill did not fill).

Other depression-filling algorithms, including the flood-fill, carving, and combined methods we have discussed, consider all depressions to be spurious. FlowFill represents the other end-member: a method that considers all depressions to be real. A perfect method would be able to reliably differentiate between the two; perfect data would only contain real depressions. Since we have neither, we inherently assume that FlowFill successfully fills spurious depressions while recognising that this may not always be true and that a user should be vigilant for any obvious errors.

**Comment**: I wonder whether the objective of determining the amount of water in each topographic sink given a runoff volume can be obtained more easily.

**Response**: It certainly can – at least, more easily with respect to a user's computational time. This is something that we are still working on and hope to share with the community in parts 2 and 3 of our series. The method we are working on should be
much more computationally efficient, but is significantly more complex to create and explain. Despite the fact that we are working toward a better way of dealing with this particular problem, we still thought it important to publish this discussion of FlowFill for a number of reasons:

1) It is important to highlight this issue in DEM preprocessing and move forward discussion about it. We hope that others will also think more about this issue and potential solutions.

2) While FlowFill is not computationally efficient, it is an intuitive method that was our first step in dealing with this particular problem.

3) There are cases (in other applications) where FlowFill may be more useful than methods of the type you have suggested. For example, since FlowFill actually routes water across the land surface, it can be used to see how flow directions and water distribution on a landscape change through time. This was not the main focus of our discussion here and is closer to what you have mentioned as FlowFill being a simplified simulation tool that models water flow. FlowFill can be used as a cellular automaton with a variety of possibilities. A user can view outputs after a given amount of iterations, or at multiple intervals after a set number of iterations each time. This could provide a simplified simulation of how flood water moves across the landscape through time. A user could also view outputs from adjacent iterations to see how dominant flow directions change through time. These or other cell-based applications would not be possible with approaches that develop flow topology based on depressions, such as the approach proposed by the reviewer.

We will mention these alternative applications in the final version of the paper.

**Comment**: I think that the paper would benefit from placing more emphasis of FlowFill being a modelling tool to study hydrological connectivity of complex landscapes, rather than a preprocessing technique.

**Response**: Thank you for your comment: our intention was to focus on one specific application of FlowFill. This ties back to our answer to the question above, where we have mentioned some other possible applications of this tool. We will mention these in the final version of the paper, although we do think that the preprocessing application of FlowFill is a significant one.

Smaller specific line and figure comments have been dealt with as suggested, and we thank Dr. Schwanghart for his careful attention.